# Polarization-transparent silicon photonic add-drop multiplexer with wideband hitless tuneability

Francesco Morichetti [1,3 ✉], Maziyar Milanizadeh[1,3], Matteo Petrini[1], Francesco Zanetto [1], Giorgio Ferrari [1], Douglas Oliveira de Aguiar[1,2], Emanuele Guglielmi[1,2], Marco Sampietro [1] & Andrea Melloni [1]

Flexible optical networks require reconfigurable devices with operation on a wavelength range of several tens of nanometers, hitless tuneability (i.e. transparency to other channels during reconfiguration), and polarization independence. All these requirements have not been achieved yet in a single photonic integrated device and this is the reason why the potential of integrated photonics is still largely unexploited in the nodes of optical communication networks. Here we report on a fully-reconfigurable add-drop silicon photonic filter, which can be tuned well beyond the extended C-band (almost 100 nm) in a complete hitless (>35 dB channel isolation) and polarization transparent (1.2 dB polarization dependent loss) way. This achievement is the result of blended strategies applied to the design, calibration, tuning and control of the device. Transmission quality assessment on dual polarization 100 Gbit/s (QPSK) and 200 Gbit/s (16-QAM) signals demonstrates the suitability for dynamic bandwidth allocation in core networks, backhaul networks, intra- and inter-datacenter interconnects.

[1] Dipartimento di Elettronica, Informazione e Bioingegneria, Milano, Italy. [2] Present address: PhotonPath s.r.l., Milano, Italy. [3] These authors contributed equally: Francesco Morichetti, Maziyar Milanizadeh. ✉email: francesco.morichetti@polimi.it

Dynamic bandwidth allocation of wavelength division multiplexing (WDM) channels among reconfigurable nodes is a key enabler for capacity growth and management of core networks, 5 G backhaul networks, intra- and inter-datacenter interconnects[1,2]. For more than two decades integrated photonics has been proposed as a promising technology for realization of compact, low cost, low-power consumption tuneable WDM filters[3–7]. In particular, coupled microring resonator (MRR) architectures fabricated on high-index-contrast platforms, such as silicon photonics, can provide good spectral performances in terms of wide passbands (several tens of GHz), steep roll-offs, and high extinction ratios (> 50 dB)[8–10]. On paper, these filters can also fulfill three main requirements which are fundamental to bring them from lab experiments to real applications: (i) operation and tuneability across a wavelength range of several tens of nanometers, matching for instance the gain bandwidth of semiconductor and fiber amplifiers; (ii) possibility of dynamically re-routing selected subsets of channels (i.e., wavelengths), while keeping full transparency for all the other channels transmitted through the device, this feature being typically referred to as "hitless" tuneability; and (iii) insensitivity to the polarization state of the input light signals, that translates into a low polarization dependent loss (PDL) and low polarization dependent crosstalk.

Actually, we say "on paper" because these requirements have been achieved only individually by some device concepts reported in the literature, but no devices have been ever demonstrated satisfying them all. For instance, wide wavelength range operation (up to about 40 nm[11,12]) was reported in silicon MRR filters designed according to Vernier schemes[13,14], that is by cascading resonators with different free spectral ranges (FSRs). Polarization independence was demonstrated in silicon nitride MRR filters by using a polarization diversity scheme[9], but without hitless tuning functionality. Only a few filter architectures were proposed that successfully implement hitless tuneability: in refs. [15,16] tuneable couplers are used as switches to disconnect the filter from the input bus waveguide, yet this approach does not apply to broadband Vernier schemes because of the appearance of off-band notches during tuning operations; in refs. [17–19] a Mach–Zehnder interferometer (MZI) bypass switch is exploited to transfer the entire input channel grid to an auxiliary filter bank, but this requires duplication of the entire filter architecture together with its control electronics.

In this work, we report on an ultra-wide-band polarization-insensitive hitless tunable filter providing a positive answer to all the above-mentioned requirements. Noteworthy, the solution we propose does not come from a customized technology process, but it can be implemented on standard silicon photonic platforms. Ultra-wide-band operation is achieved through a modified Vernier scheme based on non-integer ratios between the FSR of the MRRs, resulting in a non-periodic, theoretically FSR-free, frequency response of the overall filter. Hitless tuning is operated by introducing controllable loss in the MRRs of the filter through the use of p-i-n junctions acting as fast (ns time-scale) variable optical attenuators (VOAs). Resonance-enhanced loss is exploited to intentionally cancel out the passband of the filter with negligible impact on the off-band response. The proposed filter concept is embedded in a polarization diversity scheme demonstrating polarization-insensitive single passband filter with hitless tunability across a wavelength range of about 100 nm (1520–1610 nm, limited by experimental equipment).

## Results

**FSR-free filter**. Figure 1a illustrates the concept of the filter architecture, which consists of a chain of directly-coupled MRRs connected to two bus waveguides via tunable couplers implemented by means of MZIs. The MRRs have suitably different radii in order to realize a Vernier scheme[13] but, with respect to conventional Vernier-based filters, non-integer ratios between the FSR of the MRRs are used to cancel out the periodicity of the frequency response of the overall filter (FSR-free response)[20]. The design procedure starts from an integer-ratios Vernier master filter[21] matching the pass-band frequency response specifications and a reasonably large FSRs. Then, the radii $R_i$ and the coupling coefficients $K_i$ of the MRRs are optimized starting from the nominal design according to a numerical procedure described in Supplementary Sec. 1 in order to achieve FSR-free frequency response, while keeping the spectral shape of the main passband over the broadest wavelength range. When the filter pass-band is tuned at different wavelengths, the power split ratio of the input/output MZIs is adapted in order to counteract the wavelength dependence of the inner directional couplers and guarantee the best impedance matching from the bus waveguides to the filter[21,22]. Non-periodic frequency response with a single-pass-band characteristic spanning across more than 120 THz (1 μm wavelength range) can be theoretically achieved (Supplementary Fig. 1), while keeping the MRR bending radius well above the lossless regime for silicon waveguides ($R_i > 7$ μm)[23].

Figure 1b shows a microphotograph of a 4 MRR filter fabricated on a 220-nm commercial silicon platform[24] (see Methods for details on the filter design parameters). The rib-shaped waveguide has a width of 500 nm and a 90-nm thick lateral slab. Figure 1c shows the measured frequency response of the filter at three wavelengths around 1538 nm. The Drop port response exhibits a 3 dB bandwidth of about 41 GHz, with less than 1 dB in-band loss and about 0.5 dB in-band ripple (evaluated across a 25 GHz pass-band). Off-band isolation is 20 dB at 33 GHz from the center of the passband and more than 30 dB at 50 GHz. At the Through port, in-band isolation averaged across 20 GHz around the center of the passband is 17.2 dB. Figure 1d shows seven selected tuning states of the filter across the maximum wavelength range observable with our measurement equipment (90 nm). Results prove that the filter exhibits an FSR-free behavior with no evidence of wavelength periodicity. Details of the passband at selected wavelengths are shown in Fig. 1(d1-d4). Due to the wavelength dispersion of the coupling coefficients $K_i$ (see Supplementary Sec. 2. "Directional coupler design"), the filter bandwidth increases from 37 GHz @1528.9 nm ($d_1$) to 53 GHz @1600.9 nm ($d_4$) while maintains good performance in terms of in-band Drop-port insertion loss (<1 dB) and Through port in band isolation (>16 dB). Off-band transmission peaks at the Drop port are rejected by more than 33 dB over the entire wavelength span, while at the Through port only tiny notches (<1.2 dB deep) sporadically appear that are originated by the cavity-enhanced round-trip loss of the first MRR of the filter (< 0.02 dB, see Supplementary Sec. 4).

**Loss-mediated hitless tuning**. Hitless tuning of the filter exploits controllable loss induced through VOAs integrated in the waveguide of the MRRs in addition to the thermal tuner[20], as shown Fig. 2a. The VOA waveguide section is realized by p-doping ($10^{20}$ cm$^{-3}$) and n-doping ($10^{20}$ cm$^{-3}$) the lateral slab at a distance of 900 nm from the waveguide core in order to introduce negligible additional round-trip loss when no voltage is applied to the p-i-n junction (see Supplementary Sec. 4 "VOAs integrated in silicon MRRs"). By forward biasing the VOA, free carriers are injected in the waveguide core and the associated cavity-enhanced loss is exploited to inhibit transmission up to complete disconnection of the filter. Injection of free carriers in silicon MRR filters, driven by either optical[25] and electrical[26] control, was successfully exploited

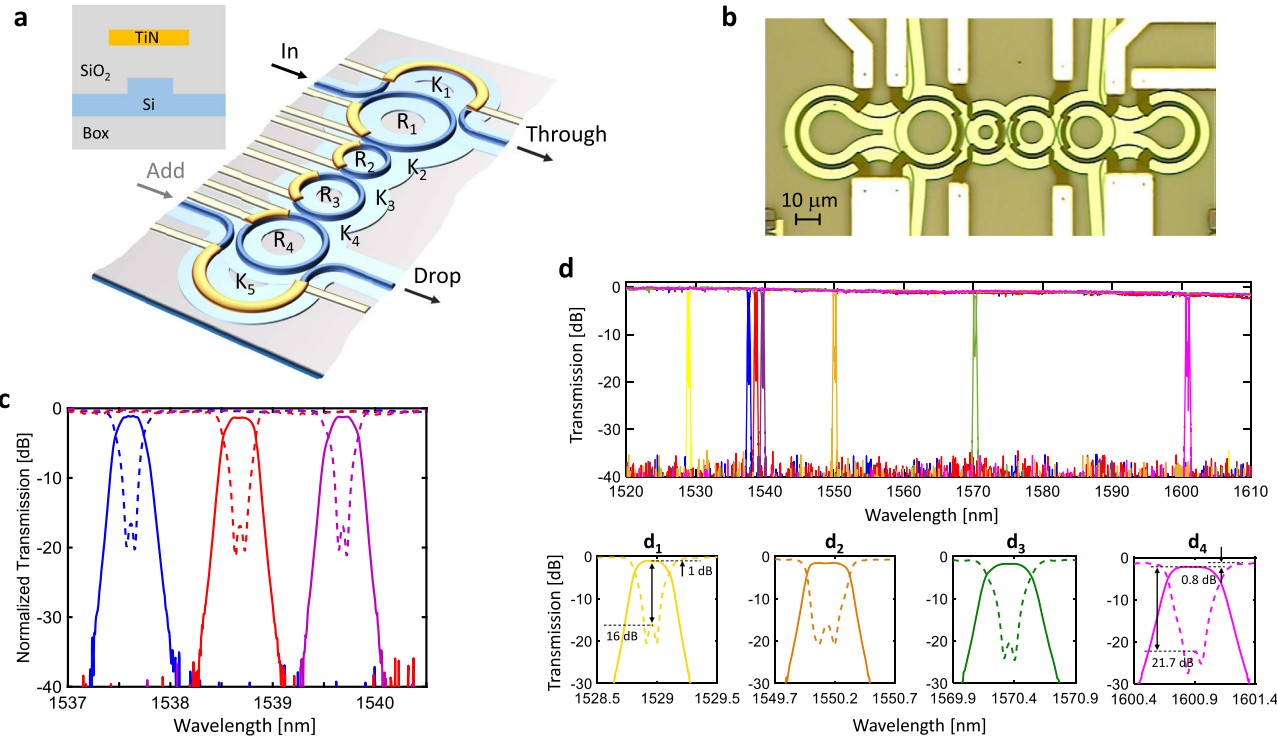

**Fig. 1 FSR-free filter. a** Schematic of the 4th order MRR Vernier filter with MZI tunable couplers. TiN heaters, controlled individually, are placed 700 nm above the waveguide core; **b** microscopic picture of fabricated device; **c** Drop (solid) and Through (dashed) port transfer function of the filter tuned at 3 channels around 1538 nm. The filter has about 40 GHz bandwidth with more than 30 dB rejection at 50 GHz from the center of the passband. **d** Frequency response of the filter tuned to 7 different channels across a 90-nm-wide wavelength range. FSR-free operation is observed for all the tuning configurations. Across the entire wavelength range, the in-band insertion loss at the Drop port (with respect to the off-band Through port transmission) is <1 dB and the in-band isolation at the Through port (with respect to the in-band Drop port transmission) is higher than 16 dB.

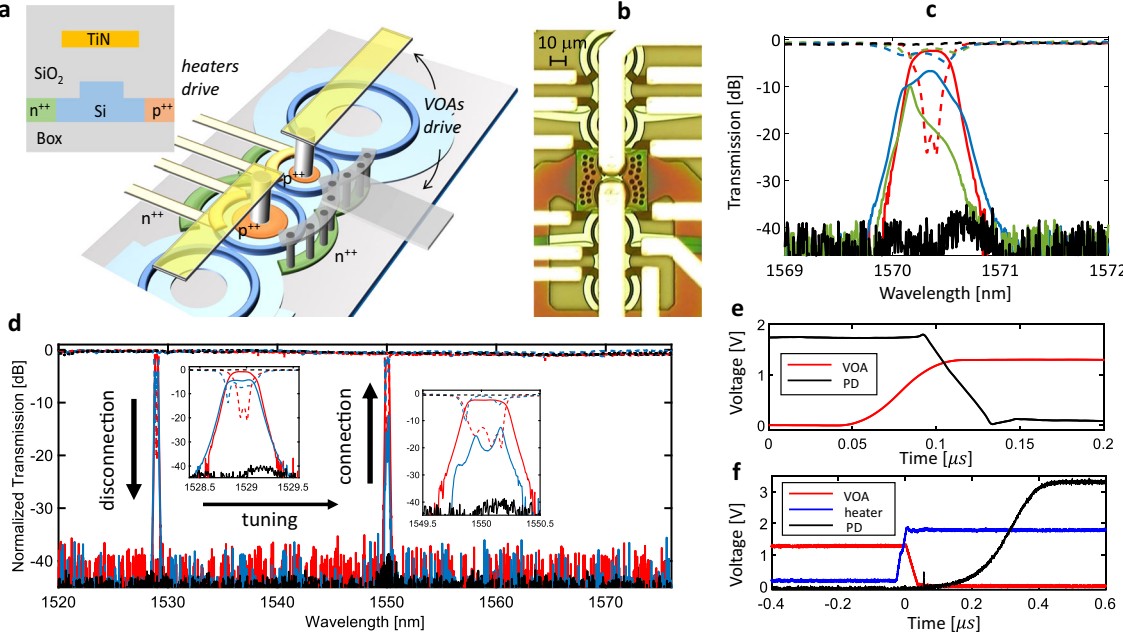

**Fig. 2 Hitless tuning. a** Scheme of a coupled MRR filter with integrated p-i-n junctions acting as VOAs in addition to integrated thermal actuators. The rib-waveguide cross section with the TiN heater and the doped regions is shown, whose dimensions are given in Supplementary Sec. 4. **b** Microscopic picture of the central section of the silicon photonic filter with integrated VOAs. **c** Measured spectral response of the filter at the Through (dashed) and Drop (solid) ports during the hitless operation at 1570.4 nm for different VOAs driving voltages: connected state (0 V, red), complete disconnected state (1.3 V, black), intermediate states (0.9 V, blue; 1 V, green). **d** Hitless reconfiguration of the filter from channel 60 (1529.55 nm) to channel 34 (1550.12 nm) of the ITU-T grid performed without introducing perturbation to WDM channels in between. Time response of the VOA (**e**) turning-on process (loss increase, filter disconnection) and (**f**) turning-off (loss decrease, filter connection), improved by heater thermal compensation. (PD = photodetector connected to the output port).

to realize fast optical switches to achieve hitless tuning of a filter, as demonstrated in ref. [20]. It is worth pointing out that hitless filter disconnection cannot be operated by increasing the loss of the first MRR of the filter, because this will make the resonator pass through the critical coupling condition[27], resulting in deep off-band notches in the Through port (see Supplementary Sec. 5). Quantitatively, for the architecture reported in Fig. 1 numerical simulations indicate that a round-loss increase by only 1.2 dB in the two inner MRRs is sufficient to introduce more than 30 dB of transmission suppression, achievable with a very low current injection (see Supplementary Sec. 5).

Figure 2b shows a top-view photograph of an FSR-free filter with integrated VOAs. All the design parameters of the filter ($K_i$, $R_i$) are the same as in the device of Fig. 1 (see Methods). The concept of loss-mediated filter disconnection is experimentally demonstrated in Fig. 2c. In the initial configuration, both VOAs are in the transparent state (no voltage applied) and the filter passband is tuned at 1570.35 nm (red curves). From this condition an increasing forward-bias voltage of 0.9 V (0.13 dB round-trip loss, blue-dashed curves), 1 V (0.18 dB loss, green-solid) and 1.3 V (0.5 dB loss, black curves) is applied to both VOAs. At the end of the process, the filter passband is almost completely suppressed, with more than 35 dB isolation at the Drop port and less than 0.3 dB ripple at the Through port.

Hitless wavelength selection is demonstrated in Fig. 2d showing an example of filter tuning from a 1529 nm to 1550 nm. Filter disconnection can be operated very fast, thanks to the ns-timescale response of carrier-injection in the VOAs. In our device, where the VOA design and the electronic driver were not optimized for fast response, we achieved a VOA turn-on time (filter disconnection) of about 80 ns (see Fig. 2e), as measured by the power suppression at a photodiode coupled to the Drop port (black curve). Once the filter is disconnected from the bus waveguide, the final wavelength (1550 nm) can be selected by driving the MRRs heaters according to a pre-calibrated look-up table[28]. High isolation enables to perform the MRR resonance detuning with no effects on the Through port transmission of the filter across the entire wavelength range (1520–1580 nm). Reconnection of the filter to the bus waveguide is a critical process because carrier-injection in the VOA action is also associated with plasma-dispersion[29] and thermal[30] effects, modifying the refractive index of the MRR waveguide. The resonance spread due these side-effects can be thermally counteracted by means of the integrated thermal tuners. In order to do that, the VOA turning off ("filter connection") needs to be conducted at a slower speed in order to match the higher time response of thermal tuners (tens of microseconds). As shown in Fig. 1f, by synchronizing the driving voltage signals of the thermal tuner (blue curve) and of VOA (red curve), the overall time response of the connection process can be as fast as 400 ns (black curve), that is more than one order of magnitude less than the heater time response (details in Supplementary Sec. 6).

**Polarization independent Add-Drop filter**. Silicon photonic waveguides and devices characteristics strongly depend on the state of polarization of the light and hence the two input orthogonal polarizations have to be treated separately. Polarization insensitivity is achieved by using the polarization diversity scheme of Fig. 3a. A polarization splitter and rotator (PSR)[24] splits the incoming signal in its two orthogonal polarizations (TE and TM) and rotates the TM polarization in order to have only TE mode propagating in all the waveguides of the circuit. The two filters are nominally identical and designed to work on the TE polarization state[9]. To finely compensate for unequal loss in the two paths, a p-i-n VOA is integrated in both arms of the

polarization diversity scheme. At the output, a polarization rotator and combiner (PRC)[24] is used to rotate from TE to TM the mode that was not rotated by the PSR, and to combine the two orthogonal modes at the output.

Figure 3b shows a photograph of the polarization diversity filter. Polarization independence was demonstrated by scrambling the polarization state of the light at the input of the device during the wavelength sweep (see Methods and Supplementary Sec. 7 "Experimental setup for polarization diversity filter"). Figure 3c shows ten independent measurements of the device spectral response, where the polarization state is randomly changed during data acquisition with a wavelength step of 1 pm (every point in each trace corresponds to a random polarization state). Results demonstrate that, when both filters are tuned, the polarization dependence of the filter is extremely small (in contrast, Supplementary Fig. 10c shows that a large PDL is introduced when only one filter is tuned). Off-band rejection at the Drop port is higher than 40 dB and no notches appear in the Through port transmission ($c_1$). Likewise, for the in-band behavior or the filter ($c_2$), in-band isolation remains higher than 17 dB, with less than 1.6 dB variation in the Drop port transmission. These power fluctuations are partially due to the experimental setup, which introduces a polarization dependent loss (PDL) of about 0.4 dB, so that the filter PDL is ultimately 1.2 dB. Panels $c_3$–$c_5$ show that, when the input polarization state is scrambled, the spread of the phase response (< 0.05 rad std), of the group delay (< 1 ps std) and of the dispersion (< 20 ps/nm std) at the Drop port of the filter is extremely low, thus confirming an almost polarization independent response of the filter architecture. Moreover, since the two filters have a very similar group delay characteristic and the scheme of Fig. 3a is geometrically balanced, polarization mode dispersion is negligible as well.

Finally, we assessed the polarization independent filter architecture from a system perspective by measuring the Bit-Error-Rate (BER) performance on a 100 Gbit/s DP-QPSK signal and a 200 Gbit/s DP-16QAM signal. A coherent commercial transceiver enabled by a real time digital signal processor (DSP) running a Soft-Decision Forward Error Correction (SD-FEC) algorithm has been used to emulate a realistic application condition. The pre-FEC BER performance for both dual-polarization signals versus the optical signal to noise ratio (OSNR) are reported in Fig. 4a, b. With reference to the scheme of Fig. 3a, the In-Through curve (blue circles) is measured with the filter tuned far apart from the signal central wavelength and confirms the negligible impact of the PSR-PRC and fiber-waveguide edge couplers, demonstrating the polarization transparency. The performance is almost indistinguishable from the one with the transmitter and receiver connected by a simple fiber loop in back to back (B2B, black diamonds). Negligible impact of the chromatic dispersion given by the first MRR was observed on signals transmitted at random carrier wavelengths along the off-band wavelength range of the filter (see Supplementary Sec. 8 "Impact of out-of-band chromatic dispersion"). The filter impact is negligible or even beneficial also for the Input-to-Drop path (orange squares), for both 100 Gbit/s and 200 Gbit/s signals. When an Add signal is launched in the Add port, the coexistence of both signals can generate at the Drop and Through ports a coherent interference, i.e. a crosstalk, fully acceptable for the 100 Gbit/s signal but more detrimental for the 200 Gbit/s, in which case it would be more appropriate a filter with higher rejection or even an add-after-drop scheme. It is worth noting that the In-Drop BER curve (purple triangles) corresponds to the Add-Through one (red triangles), assessing the symmetric behavior of the architecture despite the Vernier schemes adopted for the filters (see Supplementary Fig. 10b). Both curves match

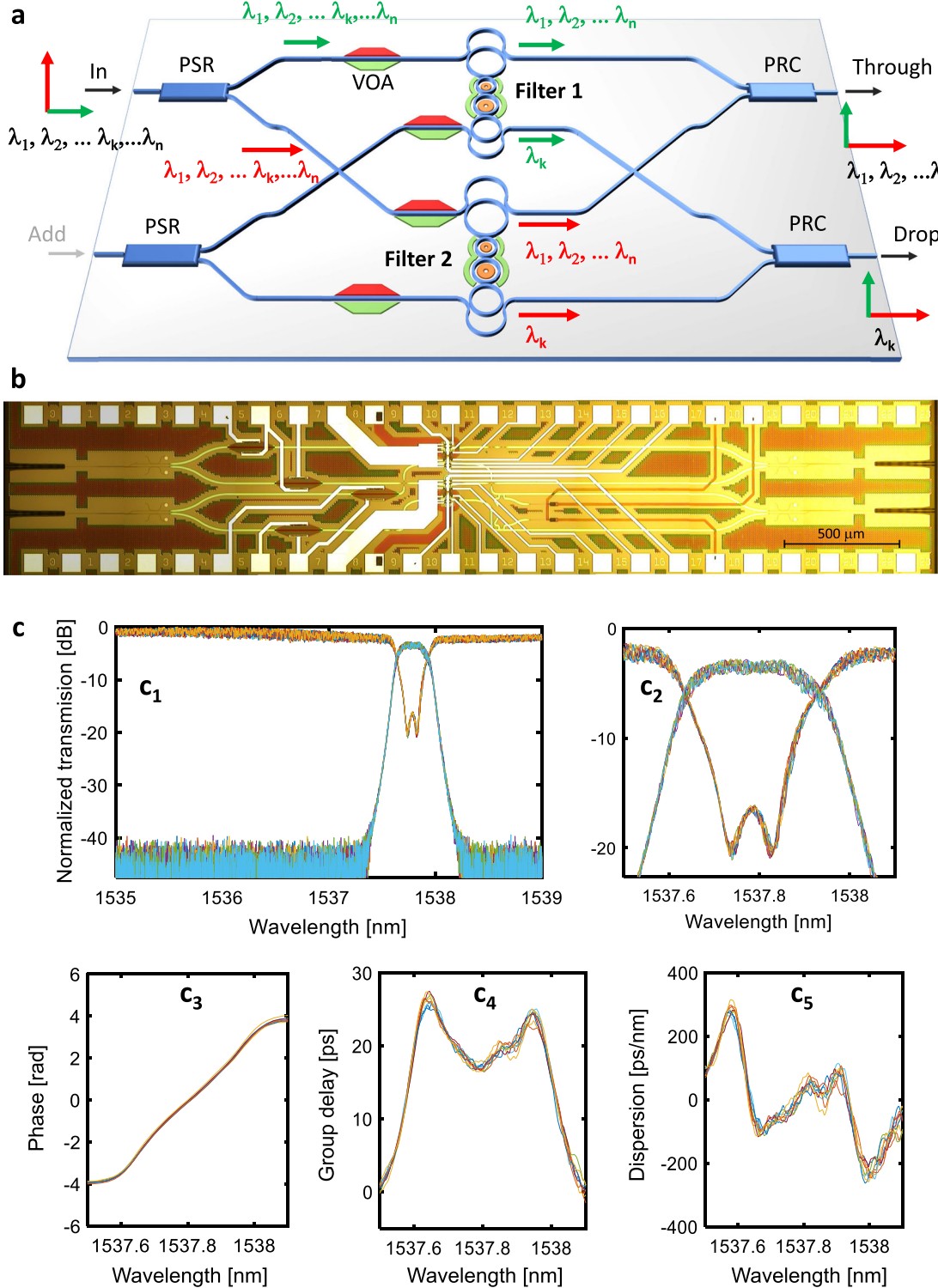

**Fig. 3 Polarization independent filter. a** Two identical hitless Vernier filters are integrated in a balanced polarization diversity scheme including polarization splitters and rotators (PSRs) at the input and polarization rotators and combiners (PRCs) at the outputs. Both filters operate on the TE mode and VOAs are used for tuning and calibration; **b** Top view microphotograph of the whole filter; **c** Drop and Through ports spectral response measured with a polarization scrambler at the input. Each point of the curves is a different polarization. Details of intensity ($c_2$), phase ($c_3$), group delay ($c_4$) and dispersion ($c_5$) of the drop port of the filter show the extremely low polarization dependence.

the B2B transceiver BER performance with an induced crosstalk equal to the in-band rejection of the filter ($-19$ dB, dashed lines) and for both signal rates the FEC of the transceiver can recover an error free condition when the curve is below the 2e-2 FEC threshold (see Supplementary Sec. 7 and 8 for details).

## Discussion

We demonstrated that a functionality strongly craved for years, that is an integrated polarization-independent filter with ultra-wide-band hitless tuneability, is nowadays realistic and fully compliant with current and future technologies for core networks

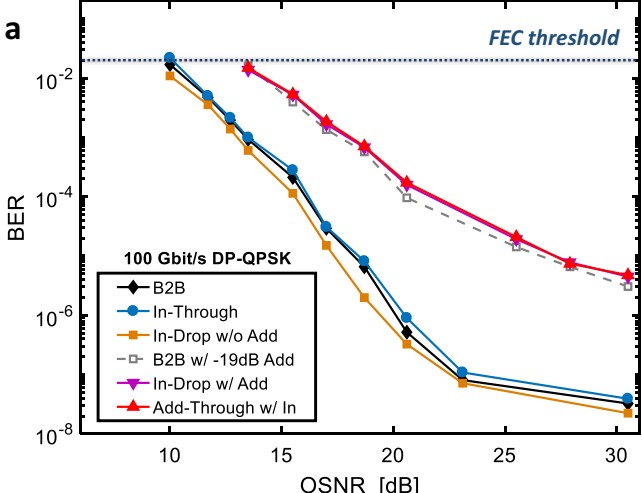

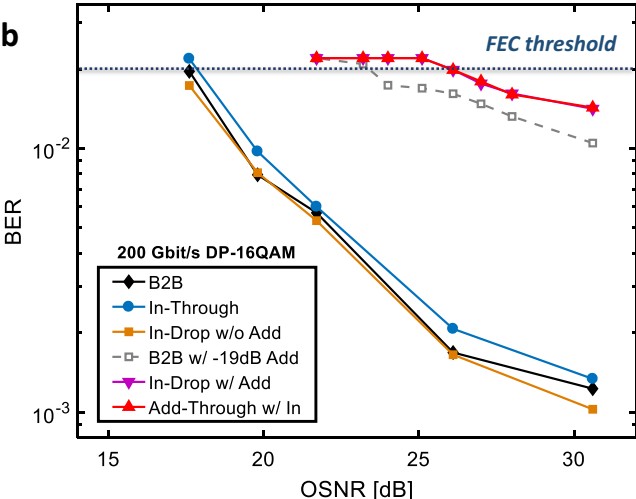

**Fig. 4 Assessment of data transmission quality.** Pre-FEC BER versus OSNR of (**a**) 100 Gbit/s DP QPSK channel and (**b**) 200 Gbit/s DP- 16QAM channel measured in the operating conditions specified in the legends. The OSNR is evaluated as the ratio between the optical signal power and the noise measured over a band of 0.1 nm close to the signal spectrum. Port naming refers to the polarization independent filter of Fig. 3a. BERs below the indicated FEC threshold are corrected in real time to an error-free condition by the FEC embedded in the receiver. In all the BER measurements, the polarization state of the light at the input of the filter is scrambled in sub-millisecond time scale.

and the emerging 5 G backhaul networks and datacenter interconnects. Advantageously, the enabling innovation is not a specific technology process, since the device was fabricated in a commercial silicon foundry. We exploited instead new strategies for the design, calibration, tuning and control of the photonic integrated circuit, which can be ported to other circuits and optical functionalities[28]. We expect that, beyond the high performance of the presented device for specific applications, these results may contribute to change the perception of what can be done now with integrated photonics, opening the door to new ultracompact, low cost, low-power consumption, fully reconfigurable and programmable photonic devices[31].

## Methods

**Filter design parameters.** The filter is designed to have a 40 GHz passband around a wavelength of 1550 nm. The radii of the four MRRs are $R_i = [14.6; 8.4;$

10.2; 12.2] μm, resulting in a FSR of [0.837; 1.454; 1.2; 1.0] THz, respectively. The power split ratios $K_i = [(7.6\% \ 7.6\%), 1\%, 0.35\%, 0.85\%, (6.3\% \ 6.3\%)]$ $(i = 1, 2, \ldots 5)$ of the directional couplers of the MRRs are optimized by changing the waveguide gap distance from $g = 200$ nm to about 530 nm (in the first and the last MRR, the MZI tuneable couplers provide up to 28.1% and 23.6% power coupling with the bus waveguides). The tuneability of the MZI couplers is exploited to counteract the $K_i$ wavelength dependence which increases versus $g$, with a 0.5%/nm slope for the outer directional couplers ($g = 200$ nm) and a 0.9%/nm slope for the inner couplers ($g = 530$ nm) (see Supplementary Sec. 2. "Directional coupler design"). The waveguide crossings of the polarization diversity scheme introduce an excess loss <0.06 dB and an optical crosstalk < −40 dB on the crossed waveguide, with no evidence of polarization rotation effects (data inferred from measurements on specific test structures). A pair of dummy crossings are introduced in the upper and lower polarization arms of the filter architecture of Fig. 3a to mitigate the effects of the crossing excess loss on the overall PDL of the filter.

**Filter tuning.** The MRR are thermally tuned by means of TiN resistive heaters (480 Ω) deposited on top of the waveguide surface at a distance of 700 nm. Thermal phase shifters enable the tuning of each MRR over more than one FSR so as to provide full filter reconfiguration as well as to compensate for native resonance spread induced by fabrication tolerances. The filter is tuned automatically by adapting its shape to match the spectrum of a 100 Gbit/s QPSK channel (28 GHz bandwidth), whose carrier frequency sets the center of the filter pass-band, according to the technique described in ref. [28]. Mitigation of thermal crosstalk among thermal tuners is performed as described in ref. [32] to improve the convergence time and accuracy of the automated tuning algorithm. The VOAs introduced in the polarization diversity scheme can be used to equalize the optical power in the two polarizations. The phase response of Fig. 3(c3) is achieved by applying Hilbert transform[33] to the frequency domain transmission curves of Fig. 3 (c2) while the group delay (c4) and dispersion (c5) are derived by subsequent differentiation versus frequency.

**Loss control through VOAs.** The forward-bias voltage driving the VOAs is applied to the p-doped regions inside the resonators through two independent metal electrodes. The n-doped regions lie at the outer side of the MRRs waveguide and are grounded to the same electrical line. The two identical VOAs have an active length of 30 μm (covering each about 60% of the MRR circumference) and introduce up to 4 dB additional round trip loss when fed with a driving current of 25.4 mA (3 V). A TiN heater 700 nm above the core guarantees both the MRR tuning and the thermal compensation of the VOA during the hitless operation (see Supplementary Secs. 4–6 for more details).

**Filter quality assessment.** The wavelength response of the filter is measured by using a tunable laser source (TLS) operating in the 1520–1610 nm wavelength range, which is synchronized with an optical spectrum analyzer (OSA). Fiber-to-chip coupling is performed by using optimized mode adapters (suspended tapers)[34] and small core fibers with a mode field diameter (MFD) of 3.2 μm, providing a coupling loss of 3 dB/facet (red curve), with <0.2 dB wavelength dependent loss. The average insertion loss of the polarization diversity scheme measured across a 60 nm wavelength range around the extended C band is about 7 dB (see Supplementary Sec. 3). The total fiber-to-fiber loss can be potentially reduced to less than 5 dB by using state-of-the art fiber-to-waveguide optical interface providing <2 dB loss across a wavelength range of 100 nm[35]. To measure the filter PDL, the polarization of the light was randomly scrambled in sub-millisecond time scale during the acquisition of the filter spectrum as well as during BER measurements. BER were performed by using a commercial transceiver generating a 100 Gbit/s double-polarization QPSK signal and a 200 Gbit/s double-polarization 16-QAM signal. Both signals have a bandwidth of 32 GHz and can be tuned along the C-band according to the 50-GHz spacing ITU-T grid. The coherent receiver assisted by a digital signal processor (DSP) can compensate chromatic dispersion (CD) up to 40,000 ps/nm (100 Gbit/s signal) or 10,000 ps/nm (200 Gbit/s signal), polarization dependent loss (PDL) up to 3 dB, polarization mode dispersion (PMD) of 15 ps. Furthermore, it can track changes of the state of polarization (SOP) of the light of 300 krad/s. The interfering signal is a 100 Gbit/s double-polarization QPSK signal (28 GHz bandwidth) generated by a different transceiver (more details in Supplementary Sec. 7).

## Data availability

The data that support the plots within this paper and other findings of this study are available without restrictions at https://doi.org/10.5281/zenodo.4588332.

## Code availability

The code used for the filter design is available from the corresponding authors on reasonable request.

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

## Acknowledgements

The authors acknowledge the staff of Polifab (www.polifab.polimi.it), the microelectronic facility of Politecnico di Milano, for the support in the assembly of photonic chips; Gaetano Bellanca for the contribution in the simulations of some photonic building blocks of the filter architecture; Alessandro Brugnoni for the support in the optical characterization activity, and Jabil Photonics for providing the optical transceiver employed for data transmission testing. The work has been supported through H2020 grant number 871658 (Nebula).

## Author contributions

F.M. and A.M conceived the filter architecture, designed the experiments and supervised the work. M.M. and M.P. designed the filter architecture, implemented the tuning scheme, and performed the optical experiments. D.A. contributed to the GDS layout of the photonic chip. F.Z., E.G. and G.F. designed the electronic system for the control of the filter. M.S. supervised the implementation of the electronic platform. F.M., A.M., M. M. and M.P. analyzed the data and wrote the manuscript.

## Competing interests

The authors declare no conflict of interest.
