## [Peer Review File · Nature Communications]

Reviewers' Comments:

Reviewer #1:

Remarks to the Author:

The paper reports a silicon photonic implementation of a reconfigurable add/drop wavelength multiplexer, which is one of the core components of WDM optical networks. In particular, it is shown that the circuit could achieve wideband tunability, hitless tuning, and polarization insensitivity. While each of these operations has been demonstrated separately before, this work shows that it is possible to achieve all three simultaneously in a single circuit, which is what would be needed for real-world operations. The overall performance of the circuit is quite impressive, representing state-of-the-art achievement in silicon photonic technology for reconfigurable wavelength multiplexer applications. The loss-mediated hitless tuning scheme is particularly innovative. Taken overall, the work is significant in that it demonstrates the viability of silicon photonics as a competitive technology to conventional fiber-based components for optical communication systems.

I recommend the paper be accepted for publication with suggestions for the authors to provide further clarifications on the following points:

- Could the authors comment on the total loss of the circuit?
- One of the main issues with silicon photonics technology compared to fiber technology is the high insertion loss due to fiber-to-chip coupling. Could the authors comment on the impact of the fiber-to-chip coupling on the overall loss and polarization insensitivity performance of the circuit?
- In the hitless tuning scheme, the first ring was not disconnected. Normally this would introduce a dip in the through port due to the allpass resonance of the first ring (even when no extra loss is introduced via a VOA). Could the authors comment on the reason why such a sharp dip was not observed for their filter?
- It would be informative to provide an estimate of the total power requirement for the circuit operation, i.e. the tuning process.

Reviewer #2:

Remarks to the Author:

The authors demonstrate an add-drop silicon photonic filter that they claim is unique in three ways:

Has an FSR of 100 nm, thus can be tuned over the C-band and beyond.

Hitless

Polarization insensitivity

The filter's functionality was verified through transmitting data at 100 Gb/s QPSK and 200 Gb/s 16-QAM.

The three aforementioned features were possible due to unique strategies in design, calibration, tuning, and control of the device, where:

FSR-free response was a result of modified Vernier scheme based on non-integer ratios between the FSR of the MRRs

The hitless switching was a result of controllable loss in the MRRs of the filter through the use of fast VOAs

Polarization insensitivity is a result of implementing a polarization transparent scheme (circuit doubling with the aid of PSRs and PRCs).

While the manuscript demonstrates a robust device and promising results, the true novelty (besides the full system demonstration) resides in one two things - the FSR-free response and the hitless tunability: where up to my knowledge, a totally FSR-free response using the Vernier scheme was not demonstrated before. The hitless tunability (although) using a two-point coupled microring (and higher order filters) was proposed earlier (as referenced in [15]), however, increasing the loss of the inner rings and using a heater to compensate for the tuning speed of the VOA is interesting.

In regards to the polarization insensitivity, the presented diversity scheme was reported since 2007 (as referenced in [6]). Based on this, while the reviewer appreciates the full system demonstration (and is aware of the challenges within its control), the reviewer thinks that the FSR-

free response and hitless operation might not be a sufficient enough reason to grant it publication in Nature Communications. Despite this fact, there are other points that the authors shall elaborate on, some of which include:

In the supplementary note section 2 - directional coupler design, the authors do not specify if the coupling coefficient of the couplers is the power's or the field's. For instance, K_{1565} and K_{MZI1} in pages 4 and 5. It is not clear if those are fields of power's.

In the supplementary note section 2, page 5, what is the K_{MZI1} and K_{MZI2} ? Those were not defined before, and why do they need to be at least greater than 28.1% and 23.6%? If the authors meant that the point couplers of the tunable MZI to be at least 28.1% and 23.6%, then that would result in a top bus waveguide and add/drop waveguides power coupling to the ring of $4K(1-K^2)$, where K is the power coupling of the point couplers, which corresponds to 80% and 72% power coupling strengths, respectively. Thus, is it not clear as to how those two values were chosen.

In Figure S2c and S4a, what are the solid lines? Are they exponential fits of the data? This should be mentioned in the figure caption.

In figure S5c, what is the source of the additional loss in the MRRs spectra? Is it an on-chip device, or fiber to chip coupling losses (non-related device loss)? This is unclear.

In the supplementary document on page 6, you say that the FCD is dominant than temperature effects. Above what voltage do you expect the temperature effects to kick in?

Formatting:

The equations are not numbered, making it hard for the reviewer to refer to certain equations - I used page numbers and paragraph locations instead of equation numbers.

There are some typos, such as at the bottom of page 6, did you mean Fig 2e? What is the MRRs mentioned in page 5, and supplementary section 3? Do you mean MRR?

Some variables are not defined.

With the above caveats, this seems like a fine paper to publish in nature comms.

Reviewer #3:

Remarks to the Author:

Overview

First, I would like to say: excellent work! The supplement also provides for a comprehensive presentation and is helpful to anyone trying to reproduce the result. My main struggle in this review was that many of the building blocks were established more than a decade ago and it wasn't completely clear that the integration of functionality provided sufficient novelty to be of interest to a wide readership, such as the one sought in a publication like nature communications. This concern is highlighted by some of the integrated devices being pulled from the foundries' PDK and the performance, even if excellent for an academic demo, not being fully there for practical applications now seeking 400 Gbps at DP-16QAM. Again, this is excellent technical work and the contribution on the design and tuning of the four-ring device is of high interest to people in telecom and likely to people in photonics in general. The question I'm raising is if it is of interest to a wider audience, and hence if nature communication is the most appropriate readership for this work. I don't have a clear answer to this question and will defer to the to the editor and other reviewers. Below, I would like to suggest a few revisions. I would classify #1-2 as being of high importance (mandatory revision), #3 of being of medium importance (suggested), and #4 of being of low importance (preference).

What is missing

1- It is my understanding that the first ring is acting as an all-pass filter out of band introducing dispersion in other channels. A clear discussion of this with out-of-band dispersion modeling or measurement (or BER impact of 16QAM on affected through channels as the authors appear to have the equipment) cannot be disregarded in the main text. This is addressed from the loss point

of view already: "while at the Through port only tiny notches (< 1.2 dB deep) sporadically appear that are originated by the cavity-enhanced round-trip loss of the first MRR of the filter (< 0.02 dB, see Supplementary Sec. 3)." The request here is to address it from a dispersion point of view.

In addition to out-of-band, the in-band dispersion is of interest as well as it is impacted by design choices. If you are not disconnecting the first ring, you may not be taking a substantial hit in power (hence claimed hitless) but you may be taking a rapid hit in through-port dispersion during tuning, which could also impact transmission. I think a few numbers on the through dispersion at various switching states would be helpful here with a short discussion putting these in context with what typical receivers today can tolerate on the tuning/switching timescale (can they adapt in time not to lose bits?). This would better present the designs trade-offs and the pros/cons of the reported implementation and better support the hitless part of the argument.

2- A "180 degree" symmetry in the implementation of polarization transparency should, at first sight, provide better performance than the "1.6 dB variation in the Drop port transmission" reported here, especially than additional VOAs are used for balancing (not great as it add to on-chip loss). The discussion of the measurement setup contribution is not clear and would generally be expected to be calibrated out in the measurements.

3- What should be presented more explicitly

3.1 Could you please provide the on-chip loss (facet to facet loss)?

3.2 How many independent controls are required to run the device and what is the power dissipation?

3.3 What length of fiber was used in the BER measurement? (it is preferred to demo that the receiver can deal with both a good length of the fiber and the filter at once)

4- Other suggestions for clarity

Drop loss and transmission loss would be preferred to be shown as annotations in spectral response figure, if possible. It is difficult to read as is, especially on d1-d4 where the font size should be adjusted. These numbers are key and the reader should not need to read through the full article text to find.

Response to the Reviewers

MANUSCRIPT ID: NCOMMS-21-01745

JOURNAL: Nature Communications

MANUSCRIPT TITLE: Polarization-transparent silicon photonic add-drop multiplexer with wideband hitless tuneability

AUTHORS: Francesco Morichetti, Mazyar Milanizadeh, Matteo Petrini, Francesco Zanetto, Giorgio Ferrari, Douglas Oliveira de Aguiar, Emanuele Guglielmi, Marco Sampietro, and Andrea Melloni

The Authors wish to thank the Reviewers for the positive comments to the work and for the constructive remarks. Reviewers' suggestions have been accurately taken into consideration to amend and improve the clarity of the manuscript.

In the following, Reviewers' comments are written in blue, while Authors' Answers and Actions are written in black.

All the modifications in the revised manuscript have been highlighted in bold.

Response to Reviewer #1

The paper reports a silicon photonic implementation of a reconfigurable add/drop wavelength multiplexer, which is one of the core components of WDM optical networks. In particular, it is shown that the circuit could achieve wideband tunability, hitless tuning, and polarization insensitivity. While each of these operations has been demonstrated separately before, this work shows that it is possible to achieve all three simultaneously in a single circuit, which is what would be needed for real-world operations. The overall performance of the circuit is quite impressive, representing state-of-the-art achievement in silicon photonic technology for reconfigurable wavelength multiplexer applications. The loss-mediated hitless tuning scheme is particularly innovative. Taken overall, the work is significant in that it demonstrates the viability of silicon photonics as a competitive technology to conventional fiber-based components for optical communication systems.

I recommend the paper be accepted for publication with suggestions for the authors to provide further clarifications on the following points:

1) Could the authors comment on the total loss of the circuit? One of the main issues with silicon photonics technology compared to fiber technology is the high insertion loss due to fiber-to-chip coupling. Could the authors comment on the impact of the fiber-to-chip coupling on the overall loss and polarization insensitivity performance of the circuit?

In our device, fiber-to-chip coupling is performed by using optimized mode adapters (suspended tapers) provided by silicon photonic foundry (AMF) and small core fibers (UHNA7, Nufern, https://www.nufern.com/pam/optical_fibers/988/UHNA7/#) with a mode field diameter (MFD) of 3.2 μm . Blue curves in Fig. R1(a) show the insertion loss of silicon waveguides with a length of 0.5 cm and 7.1 cm across a wavelength range of 60 nm from 1520 nm to 1580 nm for TE polarized input light. From this measurement, we estimated a propagation loss of less than 1 dB/cm and a coupling loss of 3 dB/facet (red curve), with less than 0.2 dB wavelength dependent loss.

Figure R1: (a) Insertion loss of 0.5 cm and 7.1 cm long waveguides (blue curves) and coupling loss of suspended tapers coupled with small core fibers (red curve) for TE polarization. (b) Total insertion loss of the polarization diversity test structure, when the polarization state of the input light is randomly scrambled during the wavelength scan.

The total loss of the polarization diversity scheme was assessed by measuring the insertion loss of the test vehicle shown in the inset of Fig. R1(b), including the I/O mode adapters (suspended tapers), a PSR/PRC pair and 5-mm-long bus waveguide. The average insertion loss measured across a 60 nm wavelength range around the extended C band is about 7 dB. The four overlapped curves, which are measured by scrambling the polarization during the wavelength scan, show a PDL of less than ± 0.6 dB. The on chip loss of the polarization diversity scheme (PSR, PRC, propagation in the bus waveguides) is less than 1 dB in total. Thanks to the low PDL we did not observe any penalty in the reported BER measurements for any scrambled configuration of the state of polarization of the input light.

Actions on the manuscript:

- A short comment on the total loss of the circuit and on the fiber-to-chip loss has been added to the Main text (page 12, Method- "Filter quality assessment").
- The following reference has been added in the Main Text as Ref [33] (and in the Supplementary Information as reference [S5]) "*Q. Fang, J. Song, X. Luo, X. Tu, L. Jia, M. Yu and G. Lo, "Low loss fiber-to-waveguide converter with a 3-D functional taper for silicon photonics," IEEE Photonics Technology Letters, vol. 28, no. 22, pp. 2533-2536, 2016.*"
- More details on the loss assessment have been included to the Supplementary Information in an additional section S3 "Optical loss and electrical power consumption" and Figure R1 has been added as

2) In the hitless tuning scheme, the first ring was not disconnected. Normally this would introduce a dip in the through port due to the allpass resonance of the first ring (even when no extra loss is introduced via a VOA). Could the authors comment on the reason why such a sharp dip was not observed for their filter?

The absence of a pronounced dip in the Through port transmission of the first MRR is essentially due to the extremely low round trip loss. As detailed in the Supplementary section S4, the round trip loss of the MRR is as low as 0.015 dB/turn and in agreement with numerical simulations reported in Fig. S8(b), this loss causes a dip of less than 1 dB, which is hardly visible in the reported experimental results.

Figure R2: Efficiency of microheaters integrated on rib waveguides: measured resonance shift of the transfer function of (a) a MRR with no integrated VOA and (b) a MRR with integrated VOA versus the electrical power dissipated on the heater.

3) It would be informative to provide an estimate of the total power requirement for the circuit operation, i.e. the tuning process.

The efficiency of the thermal tuners integrated in the rib waveguides of the filter was assessed by considering the single-MRR test structures shown in Fig. R2. The two MRRs are realized by using the same waveguide cross section (500 nm with, 90 nm slab height) and share the same bending radius of 8.4 μm , the same directional couplers (415 nm and 535 nm gap distances of the two couplers) and the same heater geometry (measured resistance 480 Ω). The only difference between the two structures is the presence of a VOA integrated in the second MRRs, together with the required metallic lines and via holes, as shown in the GDS layout of the two devices in the upper panels of Fig. R2.

As reported Fig. R2(a), the heater integrated in the MRR with no VOA requires about 31 mW to provide a π -shift, corresponding to half FSR wavelength shift of the MRR transfer function. This result is comparable to the efficiency of similar devices reported in the literature. In the MRR with integrated VOA of Fig. R2(b), the doped-silicon slab surrounding the MRR and the metallic lines required to feed current into the VOA introduce additional thermal paths for the heat generated by the heater, causing heat dissipation outside the core of the waveguide. As a result, the electrical power required for introducing a π -shift raises to about 46 mW.

The MZI tuneable couplers of the filter need to be thermally controlled within a π phase shift, this requiring about 30 mW each.

Regarding the VOA, complete disconnection of the filter requires a round trip loss of 4 dB in the two inner MRRs. This effect is achieved when the VOA is fed with a driving current of 25 mA (3 V), resulting in a dissipated electrical power of 75 mW for each VOA.

As a result, the complete reconfiguration process of the filter requires in the worst case (2π phase shift to all the MRRs and π -shift to both MZI) a peak power dissipation of about 370 mW for the thermal tuning (124 mW for the tuning of the 1st and 4th MRR, 184 mW for the tuning of the inner MRRs, 60 mW for the control of the MZI tunable couplers) and an energy consumption of less than 15 μ J for the filter disconnection through the switching on of the two VOAs (150 mW power for less than 100 μ s).

Action on the manuscript.

All the details on the electrical power consumption required for the filter tuning have been included in the Supplementary Sec. S3 “Optical loss and electrical power consumption”.

Response to Reviewer #2

The authors demonstrate an add-drop silicon photonic filter that they claim is unique in three ways:

Has an FSR of 100 nm, thus can be tuned over the C-band and beyond.

Hitless

Polarization insensitivity

The filter's functionality was verified through transmitting data at 100 Gb/s QPSK and 200 Gb/s 16-QAM.

The three aforementioned features were possible due to unique strategies in design, calibration, tuning, and control of the device, where:

FSR-free response was a result of modified Vernier scheme based on non-integer ratios between the FSR of the MRRs

The hitless switching was a result of controllable loss in the MRRs of the filter through the use of fast VOAs

Polarization insensitivity is a result of implementing a polarization transparent scheme (circuit doubling with the aid of PSRs and PRCs).

While the manuscript demonstrates a robust device and promising results, the true novelty (besides the full system demonstration) resides in one two things - the FSR-free response and the hitless tunability: where up to my knowledge, a totally FSR-free response using the Vernier scheme was not demonstrated before. The hitless tunability (although) using a two-point coupled microring (and higher order filters) was proposed earlier (as referenced in [15]), however, increasing the loss of the inner rings and using a heater to compensate for the tuning speed of the VOA is interesting. In regard to the polarization insensitivity, the presented diversity scheme was reported since 2007 (as referenced in [6]). Based on this, while the reviewer appreciates the full system demonstration (and is aware of the challenges within its control), the reviewer thinks that the FSR-free response and hitless operation might not be a sufficient enough reason to grant it publication in Nature Communications. Despite this fact, there are other points that the authors shall elaborate on, some of which include:

In the supplementary note section 2 - directional coupler design, the authors do not specify if the coupling coefficient of the couplers is the power's or the field's. For instance, K_{1565} and K_{MZI1} in pages 4 and 5. It is not clear if those are fields of power's.

Actually, at the beginning of section S2 of the supplementary information, it is written that we refer to the power coupling K_i .

Action on the manuscript.

To avoid possible misunderstandings, the text has been amended in the following part of this section to clarify that these numbers indicate the power coupling coefficients of the directional couplers of the filter.

In the supplementary note section 2, page 5, what is the K_{MZI1} and K_{MZI2} ? Those were not defined before, and why do they need to be at least greater than 28.1% and 23.6%? If the authors meant that the point couplers of the tunable MZI to be at least 28.1% and 23.6%, then that would result in a top bus waveguide and add/drop waveguides power coupling to the ring of $4K(1-K^2)$, where K is the power coupling of the point couplers, which corresponds to 80% and 72% power coupling strengths, respectively. Thus, is it not clear as to how those two values were chosen.

We agree with the Reviewer that on this point some amendments of the text are necessary to avoid any misunderstandings.

The MZI tuneable couplers have to guarantee a controllable power coupling between the bus waveguide and the MRR up to a maximum value of 28.1% (for MZI1) and to 23.6% (for MZI2). This is achieved when the power coupling coefficients of the point couplers of the MZIs are respectively 7.6% and 6.3%.

Actions on the manuscript:

- In the Main Text (page 11, “Methods – Filter design parameters”), the following sentence has been rephrased “The power split ratios $K_i = [(7.6\% \ 7.6\%), 1\%, 0.35\%, 0.85\%, (6.3\% \ 6.3\%)]$ ($i = 1, 2, \dots 5$) of the directional couplers of the MRRs are optimized by changing the waveguide gap distance from $g = 200$ nm to about 530 nm (in the first and the last MRR, the MZI tuneable couplers provide up to 28.1% and 23.6% power coupling with the bus waveguides).”
- In the Supplementary Information (page 2, Sec. 1, “Numerical optimization of the FSR-free filter design”), the following sentence has been amended “The coupling coefficients of the master filter $K_i = [7\%, 0.8\%, 0.35\%, 0.98\%, 6\%]$ provide a passband of 40 GHz, where the power coupling ratio of the 1st MRR (7%) and the 4th MRR (6%) refer to the value of each point couplers of the MZI.” The caption of Fig. S1 has been amended accordingly.
- In the Supplementary Information (pages 4-5, Sec. 2, “Directional coupler design”), the following sentence has been amended “To achieve the required power coupling ratio $K_{MZI1} = 28.1\%$ and $K_{MZI5} = 23.6\%$ with a tuneable MZI, the two point couplers must have a power coupling ratio $K_{1,5}$ of at least

$$K_{1,5} = \sin^2 \left(0.5 \sin^{-1}(\sqrt{K_{MZI1,5}}) \right)$$

that are $K_1 = 7.6\%$ and $K_5 \geq 6.3\%$.”

In Figure S2c and S4a, what are the solid lines? Are they exponential fits of the data? This should be mentioned in the figure caption.

Yes, dashed straight lines in Figs. S2c and S4a are the exponential fits of the simulated data points. This information has been added in both captions.

In figure S5c, what is the source of the additional loss in the MRRs spectra? Is it an on-chip device, or fiber to chip coupling losses (non-related device loss)? This is unclear.

(Note: after manuscript amendment this figure is now numbered as Fig. S6).

We believe that the Reviewer refers to the insertion loss in the transmission of the Through port response of the MRRs (red curves). This is essentially due to fiber-to-chip coupling loss. For these test structures, the optical coupling was performed vertically by using grating couplers, introducing a loss of about 3-4 dB/grating.

Action on the manuscript.

A sentence has been added to Supplementary Sec. 3 (page 8) clarifying this point.

In the supplementary document on page 6, you say that the FCD is dominant than temperature effects. Above what voltage do you expect the temperature effects to kick in?

Stress tests were performed on test devices to explore the maximum attenuation that can be produced by the VOAs integrated in the MRRs without the occurrence of irreversible damaging effects. In the full range of operational voltages (up to 3 V, corresponding to a round-trip loss of 4 dB, which is high enough for the implementation of the hitless tuning of the filter), we always observed a blue-shift of the MRR spectral response demonstrating that in these devices FCD is the dominant effect.

Formatting:

The equations are not numbered, making it hard for the reviewer to refer to certain equations - I used page numbers and paragraph locations instead of equation numbers.

Numbering has been added to equations.

There are some typos, such as at the bottom of page 6, did you mean Fig 2e? What is the MMRs mentioned in page 5, and supplementary section 3? Do you mean MRR? Some variables are not defined.

We thank a lot the Reviewer for pointing out these typos in the text.

We made a fine polishing of the text to remove all the remaining typos and to define all the variables.

With the above caveats, this seems like a fine paper to publish in nature comms.

Response to Reviewer #3

Overview

First, I would like to say: excellent work! The supplement also provides for a comprehensive presentation and is helpful to anyone trying to reproduce the result. My main struggle in this review was that many of the building blocks were established more than a decade ago and it wasn't completely clear that the integration of functionality provided sufficient novelty to be of interest to a wide readership, such as the one sought in a publication like nature communications. This concern is highlighted by some of the integrated devices being pulled from the foundries' PDK and the performance, even if excellent for an academic demo, not being fully there for practical applications now seeking 400 Gbps at DP-16QAM. Again, this is excellent technical work and the contribution on the design and tuning of the four-ring device is of high interest to people in telecom and likely to people in photonics in general. The question I'm raising is if it is of interest to a wider audience, and hence if nature communication is the most appropriate readership for this work. I don't have a clear answer to this question and will defer to the to the editor and other reviewers. Below, I would like to suggest a few revisions. I would classify #1-2 as being of high importance (mandatory revision), #3 of being of medium importance (suggested), and #4 of being of low importance (preference).

What is missing

1- It is my understanding that the first ring is acting as an all-pass filter out of band introducing dispersion in other channels. A clear discussion of this with out-of-band dispersion modeling or measurement (or BER impact of 16QAM on affected through channels as the authors appear to have the equipment) cannot be disregarded in the main text. This is addressed from the loss point of view already: "while at the Through port only tiny notches (< 1.2 dB deep) sporadically appear that are originated by the cavity-enhanced round-trip loss of the first MRR of the filter (< 0.02 dB, see Supplementary Sec. 3)." The request here is to address it from a dispersion point of view.

We agree with the Reviewer that the contribution of the out-of-band all-pass filters need to be considered also from the point of view of chromatic dispersion (CD).

Numerical simulations reported in Fig. R3(a) show that the CD introduced by the out-of-band all-pass filters is less than 40 ps/nm, that is less than the CD caused by 3 km of standard fiber (17 ns/nm·km for G.652 fiber). As written in the Methods (page 13, "Filter quality assessment") the commercial transceiver that we used (Jabil Photonics CFP2-DCO) can compensate CD of up to 40000 ps/nm (100 G signal) or 10000 ps/nm (200 G signal), that is more than two orders of magnitude higher.

To prove the negligible impact of the out-of-band CD, we performed transmission experiments on signals at random carrier wavelengths along the off-band wavelength range of the filter. As shown by the BER curves reported in Fig. R3(b), no OSNR penalties due to CD effects were observed.

Action on the manuscript.

In the Supplementary Information, a section (Sec. 8 “Impact of out-of-band chromatic dispersion (CD)”) has been added specifically addressing the impact of the chromatic dispersion introduced by the first MRR on other channels. The following sentence has been added to the main text (page 9) **“Negligible impact of the chromatic dispersion given by the first MRR was observed on signals transmitted at random carrier wavelengths along the off-band wavelength range of the filter”**

Figure R3: (a) Numerical simulations of the chromatic dispersion introduced by first ring of the filter acting as an out-of-band all-pass filter; (b) BER curve of 100 Gbit/s signals transmitted at random carrier wavelengths along the off-band wavelength range of the filter.

In addition to out-of-band, the in-band dispersion is of interest as well as it is impacted by design choices. If you are not disconnecting the first ring, you may not be taking a substantial hit in power (hence claimed hitless) but you may be taking a rapid hit in through-port dispersion during tuning, which could also impact transmission. I think a few numbers on the through dispersion at various switching states would be helpful here with a short discussion putting these in context with what typical receivers today can tolerate on the tuning/switching timescale (can they adapt in time not to lose bits?). This would better present the designs trade-offs and the pros/cons of the reported implementation and better support the hitless part of the argument.

During the switching (reconfiguration of the filter) there is no data transmission at the two wavelengths corresponding to the initial state and final state of the filter. Therefore, during the switching, in-band dispersion is not of practical interest.

In-band dispersion matters in the connected state of the filter. The maximum in-band dispersion at the Drop port of the filter (about 100 ps/nm at the edges of the 3dB pass band, see Fig. 3c₅) is actually much less than the maximum CD than can be compensated by the receiver (see more details in answer 3.3).

2- A “180 degree” symmetry in the implementation of polarization transparency should, at first sight, provide better performance than the “1.6 dB variation in the Drop port transmission” reported here, especially than additional VOAs are used for balancing (not great as it add to on-chip loss). The discussion of the measurement setup contribution is not clear and would generally be expected to be calibrated out in the measurements.

We agree with this comment of the Reviewer and indeed in our device we exploited “180 degree” symmetry in the implementation of the polarization diversity scheme.

The performance of the polarization diversity scheme employed in the project was assessed by comparing the test vehicles shown in the insets of Fig. R3. Each structure consists of a polarization splitter and rotator (PSR) that is directly connected to a polarization rotator and combiner (PRC) by two straight waveguides. Both devices are standard building blocks provided by the silicon foundry. The main difference between these structures is that in structure (a) a single rotation occurs at each arm of the polarization diversity scheme, while in (b) the polarization of the field in the upper arm is rotated twice (in the input PSR and in the output PRC). Experimental results show the total insertion loss measured across a 60 nm wavelength range, when the polarization of the light coupled at the input of the PSR is randomly scrambled. For the structure (a) a polarization dependent loss (PDL) of less than +/- 0.6 dB is observed, while in (b) the measured PDL increases to more than +/- 1.5 dB.

Structure (a), minimizing the overall PDL of the polarization diversity scheme, is the one that we adopted in the design of our filter.

Figure 4. Comparison between the insertion loss of two polarization diversity test structures, when the polarization state of the input light is randomly scrambled during the wavelength scan: (a) a single rotation occurs in each arm of the polarization diversity scheme, (b) the polarization of the field in the upper arm is rotated twice (in the PSR and in the PRC).

Regarding the VOAs integrated in the polarization diversity scheme, they were designed to compensate for the occurrence of undesired power unbalance between the two arms of the circuit and to facilitate the calibration process of the filter by switching off the two paths individually. Actually, since the fabricated polarization diversity scheme is well balanced, during the operating of the filter both the VOAs are in the full transparent state and no measurable extra loss is added.

3- What should be presented more explicitly

3.1 Could you please provide the on-chip loss (facet to facet loss)?

A detailed loss analysis has been provided in the answer 1) to Reviewer #1 and the manuscript has been amended accordingly.

3.2 How many independent controls are required to run the device and what is the power dissipation?

The device is controlled by 8 independent controls, namely:

- thermal tuning of the 4 MRRs of the filter;
- thermal tuning of the 2 MZIs tuneable couplers;
- electrical control of the 2 VOAs integrated in the inner MRRs of the filter.

Detailed information on the power dissipation due to all the independent controls of the filter has been provided in the answer 3) to Reviewer 1.

Action on the manuscript.

All the details on the electrical power consumption required for the filter tuning have been included in the Supplementary Sec. S3 "Optical loss and electrical power consumption".

3.3 What length of fiber was used in the BER measurement? (it is preferred to demo that the receiver can deal with both a good length of the fiber and the filter at once)

The data transmission experiments reported in this work were achieved in back-to-back configuration using a few tens of meters of optical fiber.

As written in the answer to remark 1) the coherent receiver that we used can compensate chromatic dispersion (CD) of up to 40000 ps/nm (100 G signal) or 10000 ps/nm (200 G signal), this numbers being two orders of magnitude higher than the CD introduced by our filter (about 100 ps/nm at the edges of the 3dB pass band, see Fig. 3c₅). Considering also the low PDL (1.2 dB) and PMD (< 1 ps) introduced by the filter, we can conclude that the filter is not expected to introduce severe impairments in a long fiber communication link.

This being said, our work mainly focuses on the proposal and demonstration of a novel device concept with performance that were never before demonstrated in a single device. Results show that it is suitable for dynamic bandwidth allocation in flexible optical networks, but extensive system-level investigations of its impact in realistic scenarios are left to future dedicated works.

4- Other suggestions for clarity

Drop loss and transmission loss would be preferred to be shown as annotations in spectral response figure, if possible. It is difficult to read as is, especially on d1-d4 where the font size should be adjusted. These numbers are key and the reader should not need to read through the full article text to find.

Actions on the manuscript:

- The in-band insertion loss at the Drop port (with respect to the off-band Through port) and the in-band isolation at the Through port (with respect to the in-band Drop port) have been added in Fig- 1(d₁) and (d₄) in order to provide information on how the filter performance changes across the considered wavelength range.
- Following the suggestion of the Reviewer, a short comment has been also added in the caption of Fig. 1d to better highlight these numbers.
- Fontsize in panels d₁-d₄ has been increased in order to improve the readability of the figure.

Reviewers' Comments:

Reviewer #1:

Remarks to the Author:

The authors have revised the manuscript to address the points raised in the previous review. I am satisfied with the authors' responses to my comments and have no further points to raise with regards to the technical content of the manuscript. I think the excellent results achieved in this work could be of wide interest to both academics and industry in terms of what can be achieved with silicon photonics technology.

Reviewer #2:

Remarks to the Author:

OK to publish.

Reviewer #3:

Remarks to the Author:

Overview

As mentioned in my first review, this is excellent work with comprehensive documentation! Most of the comments have been addressed and I believe that the paper is, in my opinion, suitable for publication in nature communications.

I agree with reviewer #2 that the true novelty in this paper is on aspects of the filter itself, which is a narrower scope of contribution than the authors claim. That being said, based on clarifications received on the journal's editorial bar, I believe that even this narrower level of novelty does warrant publication in nature communications as the advances reported are highly useful to the telecom community.

Below, please find a few additional comments / clarifications that would benefit the reader if the authors have the opportunity to further refine the paper.

#1 - Chromatic dispersion

I agree with the authors that the **static** chromatic dispersion (in- and out-of-band) can be compensated by most transceivers. My concern was mainly with **dynamic** changes in chromatic dispersion during the tuning process. As the first ring is tuned without being disconnected, its resonances may temporarily overlap with adjacent channels during the tuning process (in and out of band may not make much difference here with sufficient loss on the second ring). This can create a sharp oscillation in chromatic dispersion on an adjacent channel during the tuning process. Such oscillation may be of too-short a timescale for a transceiver to correct. The transceiver's capability to deal with **transient** oscillations in chromatic dispersion is likely substantially weaker than the quoted capability to compensate for **static** chromatic dispersion. Hence, even if there is a large margin to **static** compensation, there may not be as much margin to **dynamic** compensation and could be a trade-off in this design.

It would help the reader to clarify what is meant by "random carrier wavelengths along the off-band wavelength range of the filter." One is interested in comparing the BER in adjacent channels at and next to resonances of the first ring. It is not fully clear if this is what is shown in R3(b) based on the description.

#2 - PDL

The PDL and wide spectral oscillations (assuming narrow ones are FP related) on R1(b) are not desired but speak to the AMF PDK elements, rather than to the authors contribution. As far as the authors' contribution, what I'm not clear about is the impact of waveguide crossings on the PDL. Based on fig 3(a), one polarization arm seems to have all the crossings while the other has none. This can add to the PDL in this layout choice as it appears to break the symmetry between the polarization arms.

#3 – Loss

What is still challenging for commercial applications is the full input fiber to output fiber loss. This has been plaguing silicon photonics from the start and sufficient on-chip functionality/complexity is required to offset the overhead of fiber coupling and polarization management. A quick search finds typical add-drop filters at ~1 dB drop loss and going up to 3.5 dB loss for some MEMS configurations. Those are not hitless, obviously, but it shows that the gap is still substantial. Considering best of breed components, one could anticipate bringing the fiber-to-fiber loss of the hitless switch down to 5-6 dB worst case, which may still be a substantial hit for in-line devices.

With best regards,

Tymon

Response to the Reviewers

MANUSCRIPT ID: NCOMMS-21-01745

JOURNAL: Nature Communications

MANUSCRIPT TITLE: Polarization-transparent silicon photonic add-drop multiplexer with wideband hitless tuneability

AUTHORS: Francesco Morichetti, Mazyar Milanizadeh, Matteo Petrini, Francesco Zanetto, Giorgio Ferrari, Douglas Oliveira de Aguiar, Emanuele Guglielmi, Marco Sampietro, and Andrea Melloni

The Authors wish to thank all the Reviewers for the positive comments to the work.

We agree with the additional comments provided by Reviewer #3, which have been addressed to improve the clarity of the manuscript.

In the following, Reviewers' comments are written in blue, while Authors' Answers and Actions are written in black.

All the modifications in the revised manuscript have been highlighted in bold.

Response to Reviewer #3

As mentioned in my first review, this is excellent work with comprehensive documentation! Most of the comments have been addressed and I believe that the paper is, in my opinion, suitable for publication in nature communications.

I agree with reviewer #2 that the true novelty in this paper is on aspects of the filter itself, which is a narrower scope of contribution than the authors claim. That being said, based on clarifications received on the journal's editorial bar, I believe that even this narrower level of novelty does warrant publication in nature communications as the advances reported are highly useful to the telecom community.

Below, please find a few additional comments / clarifications that would benefit the reader if the authors have the opportunity to further refine the paper.

#1 - Chromatic dispersion

I agree with the authors that the *static* chromatic dispersion (in- and out-of-band) can be compensated by most transceivers. My concern was mainly with *dynamic* changes in chromatic dispersion during the tuning process. As the first ring is tuned without being disconnected, its resonances may temporarily overlap with adjacent channels during the tuning process (in and out of band may not make much difference here with sufficient loss on the second ring). This can create a sharp oscillation in chromatic dispersion on an adjacent channel during the tuning process. Such oscillation may be of too-short a timescale for a transceiver to correct. The transceiver's capability to deal with *transient* oscillations in chromatic dispersion is likely substantially weaker than the quoted capability to compensate for *static* chromatic dispersion. Hence, even if there is a large margin to *static* compensation, there may not be as much margin to *dynamic* compensation and could be a trade-off in this design.

We agree with the Reviewer that the large margin of the transceiver in the compensation of static chromatic dispersion may not apply in the case of dynamic changes in chromatic dispersion. Actually, no specifications are provided on the capability of our transceiver to deal with transient oscillations in chromatic dispersion, so the only answer we can give to this concern comes from a direct experiment that we performed.

Referring to the case of Fig 3d [here reported for clarity as Fig. RR1(a)] we transmitted a 100 Gbit/s signal with carrier wavelength 1540.56 nm (ITU-T channel 46) and we measured the BER versus time during the switching of the filter from channel 60 (1529.55nm) to channel 34 (1550.12nm). As shown in Fig. RR1(b), once the filter is disconnected from the bus waveguide, the wavelength switch is operated after 5 s on a time scale of less than 10 μ s, as given by the time response of the thermal tuners. The transceiver provides an updated BER measurement every 1 s over a time window of 10 s. Results show no changes in the measured BER (pre-FEC) that remains below 2×10^{-8} (OSNR 30 dB) during the entire tuning process.

Therefore, we can conclude that dynamic changes in the chromatic dispersion (as well as in the loss) during the tuning process are within the dynamic compensation margin of the transceiver and are negligible for the system performance.

Action on the manuscript.

Figure RR1(b) has been added to the Supplementary Information (Sec. 8. "Impact of out-of-band chromatic dispersion") as Fig. S12 together with the comment reported above.

It would help the reader to clarify what is meant by "random carrier wavelengths along the off-band wavelength range of the filter." One is interested in comparing the BER in adjacent channels at and next to resonances of the first ring. It is not fully clear if this is what is shown in R3(b) based on the description.

We agree with the Reviewer that this sentence is a bit misleading.

BER measurements were actually performed on all the ITU-T WDM channels across the extended C-band and no significant differences were observed. BER curves in Fig. S11(b) show the results for some of the measured channels, namely the ITU-T channel 47 (1539.77 nm), 40 (1545.32 nm), 34 (1550.12 nm), 29 (1554.13 nm), and 22 (1559.79 nm). The comment to Fig. S11(b) has been rephrased to clarify this point.

Figure RR1. (a) Hitless reconfiguration of the filter from channel 60 (1529.55nm) to channel 34 (1550.12nm) of the ITU-T grid. (b) Pre-FEC BER of a 100 Gbit/s DP QPSK channel transmitted at a carrier wavelength of 1540 nm (ITU-T channel 46) during the hitless tuning of the filter.

#2 – PDL

The PDL and wide spectral oscillations (assuming narrow ones are FP related) on R1(b) are not desired but speak to the AMF PDK elements, rather than to the authors contribution. As far as the authors' contribution, what I'm not clear about is the impact of waveguide crossings on the PDL. Based on fig 3(a), one polarization arm seems to have all the crossings while the other has none. This can add to the PDL in this layout choice as it appears to break the symmetry between the polarization arms.

The Reviewer is referring to a key issue that was actually taken into consideration in the design of the filter architecture, even though it was not specifically mentioned in the manuscript.

In order to maintain the symmetry between the polarization arms and to reduce the total PDL of the circuit, two dummy crossings were inserted both in the upper arm and in the lower arm of the polarization diversity architecture shown in Fig. 3a. Here we show the GDS layout of the circuit, with a zoom around where the two waveguide sections where dummy crossings are inserted.

Action on the manuscript.

The following sentence was added to the Method Section (Filter design parameters):

“A pair of dummy crossings are introduced in the upper and lower polarization arms of the filter architecture of Fig. 3a to mitigate the effects of the crossing excess loss on the overall PDL of the filter.”

#3 – Loss

What is still challenging for commercial applications is the full input fiber to output fiber loss. This has been plaguing silicon photonics from the start and sufficient on-chip functionality/complexity is required to offset the overhead of fiber coupling and polarization management. A quick search finds typical add-drop filters at ~1 dB drop loss and going up to 3.5 dB loss for some MEMS configurations. Those are not hitless, obviously, but it shows that the gap is still substantial. Considering best of breed components, one could anticipate bringing the fiber-to-fiber loss of the hitless switch down to 5-6 dB worst case, which may still be a substantial hit for in-line devices.

We agree with the Reviewer that, by using state-of-the-art components, the total insertion loss of proposed device can be lowered from the 7-8 dB loss of our work to less than 6 dB. For instance, in this recent paper

T. Barwicz et al., "Advances in Interfacing Optical Fibers to Nanophotonic Waveguides Via Mechanically Compliant Polymer Waveguides," in IEEE Journal of Selected Topics in Quantum Electronics, vol. 26, no. 2, pp. 1-12, March-April 2020, Art no. 3700312, doi: 10.1109/JSTQE.2020.2964383.

fiber to silicon nanophotonic optical interfaces are demonstrated providing less than 2 dB loss across a 100 nm bandwidth (which is 1 dB lower than the adiabatic taper used in our work).

Action on the manuscript.

The following sentence has been added to the Method Section (Filter quality assessment):

“The total fiber-to-fiber loss can be potentially reduced to less than 5 dB by using state-of-the art fiber-to-waveguide optical interface providing less than 2 dB loss across a wavelength range of 100 nm.”

The above mentioned paper has been included in the reference list as Ref. 34.

Reviewers' Comments:

Reviewer #3:

Remarks to the Author:

I would like to thank the authors for further refinements to the paper. I believe this paper will be an excellent contribution to nature communications and will be well regarded by readers in the field.

Best,

Tymon